# Building Resilience and Competence in Bachelor Nursing Students: A Narrative Review Based on Social Cognitive Theory

**DOI:** 10.3390/nursrep15070253

**Published:** 2025-07-11

**Authors:** Elisabeth Wille, Helene Margrethe Storebø Opheim, Daisy Michelle Princeton, Sezer Kisa, Kari Jonsbu Hjerpaasen

**Affiliations:** Department of Nursing and Health Promotion, Faculty of Health Sciences, Oslo Metropolitan University, 0130 Oslo, Norway; heleneo@oslomet.no (H.M.S.O.); dapri@oslomet.no (D.M.P.); sezkis@oslomet.no (S.K.); khjerpaa@oslomet.no (K.J.H.)

**Keywords:** clinical competence, clinical education, clinical learning environment, educator behaviors, nursing students, resilience, social cognitive theory, teaching strategies

## Abstract

**Background/Objectives:** In contemporary nursing education, clinical competence and psychological resilience are both essential; however, they are often treated as separate outcomes. Clinical placements are a central component of nursing education, and often expose students to high levels of stress, emotional challenges, and complex clinical demands. Building both clinical competence and psychological resilience during this phase is crucial to preparing students for the realities of professional practice. This narrative review, grounded in Bandura’s social cognitive theory (SCT), explores how educator behaviors, teaching strategies, and learning environments interact to influence both domains in undergraduate nursing students. **Methods:** A comprehensive search was conducted in PubMed, CINAHL, and PsycINFO for peer-reviewed articles published between 1 January 2000 and 2 May 2025. Relevant studies were thematically analyzed and were guided by Bandura’s SCT constructs, including observational learning, self-efficacy, and reciprocal determinism. No formal study appraisal was conducted due to the narrative nature of the review. **Findings:** Nineteen studies were included in the review. The findings highlight that reflective practices, structured feedback, peer learning, and strategies to build self-efficacy are central to building resilience and competence. Supportive educator behaviors such as mentoring, emotional support, and professional role modeling, were strongly associated with increased student confidence, emotional regulation, and adaptability. Psychologically safe clinical learning environments further enhanced self-efficacy and active engagement. In contrast, unsupportive or inconsistent environments were linked to student stress, disengagement, and reduced performance. **Conclusions:** This review highlights the need to move beyond traditional competency-based models toward an integrated approach that equally values psychological preparedness and resilience. Findings suggest a shift toward emotionally and socially integrated clinical education.

## 1. Introduction

Preparing resilient and clinically competent nurses is an urgent priority for global healthcare systems grappling with workforce shortages, aging populations, and increasing complexity in care delivery. Clinical education plays a pivotal role in achieving this goal, bridging the gap between theoretical learning and real-world practice. However, the transition from the classroom to the clinic often exposes students to long hours, emotional stressors, and high-performance expectations that can challenge both their technical development and psychological well-being [1,2,3,4,5,6].

In this context, two complementary outcomes have emerged as essential in nursing education: clinical competence, which refers to the ability to apply technical and interpersonal skills effectively, and psychological resilience, or the capacity to adapt to adversity and maintain emotional stability under pressure. The American Psychological Association defines resilience as “the process and outcome of successfully adapting to difficult or challenging life experiences, especially through mental, emotional, and behavioral flexibility and adjustment to external and internal demands” [7]. Its importance is particularly evident in clinical practice, where students must apply theoretical knowledge in unpredictable, emotionally charged, and often high-stakes situations. This complexity demands both technical skill and emotional adaptability. Resilience helps students navigate clinical stressors, maintain well-being, and persist through the demands of training and has been linked to lower levels of burnout and a reduced intention to leave nursing programs [8]. Studies suggest that resilience and competence—traditionally treated as distinct domains—are deeply interconnected: students with higher resilience demonstrate better coping skills, greater self-efficacy, stronger academic performance, and a lower risk of burnout and attrition [8,9,10].

While some programs emphasize individual traits such as optimism or self-regulation, few provide structured pedagogical models that integrate resilience-building into teaching practices. The role of clinical educators such as models of coping, communication, and ethical conduct, is particularly underutilized, despite strong evidence that educator behaviors and learning environments profoundly shape student outcomes. Students frequently report inconsistent role modeling, a lack of psychological safety, and insufficient support during placements [11,12,13], underscoring the need for evidence-informed strategies that actively foster both competence and resilience.

This review adopts Bandura’s social cognitive theory (SCT) [14] as a guiding framework to explore how competence and resilience are codeveloped in clinical education. According to SCT, learning results from the dynamic interplay between personal factors, behaviors, and environmental influences. Central to this framework is the concept of self-efficacy, which is the belief in one’s ability to manage tasks and challenges, and is closely linked to clinical performance and psychological resilience [15,16]. Educator behaviors, observational learning, and supportive environments are key mechanisms through which students build self-efficacy and internalize professional norms.

Despite growing interest, efforts to foster resilience in clinical education often remain fragmented. While some programs emphasize individual traits such as optimism or self-regulation, few offer structured pedagogical models that intentionally integrate resilience-building strategies into clinical teaching [17]. Moreover, the role of educators as models of coping, communication, and professional behavior is often underemphasized, even though educator behaviors and the clinical environment profoundly shape students’ experiences [18,19,20]. Inconsistent role modeling, a lack of psychological safety, and insufficient support are frequently cited by students as barriers to learning and emotional growth [11,21].

This narrative review is guided by the proposition that resilience is not a fixed personal trait, but a dynamic, teachable capacity shaped by educator inputs, learning environments, and social processes. By applying SCT as a guiding framework, we aim to synthesize the evidence on how clinical educators can foster both clinical competence and psychological resilience in undergraduate nursing students. Specifically, this review addresses the following questions:What teaching strategies and educator behaviors support the development of both clinical competence and psychological resilience in nursing students?How do the characteristics of the clinical learning environment promote these outcomes?What educational processes mediate the relationship between educator inputs and the development of competence and resilience?

In this context, “educator inputs” refer to actions such as supervision, feedback, and role modeling, while “educational processes” include mechanisms like reflective practice, peer learning, and the development of self-efficacy. Through this framework, we seek to identify evidence-informed strategies that nurse educators can implement to better support students, reduce attrition, and strengthen the future nursing workforce.

## 2. Materials and Methods

### 2.1. Design

This narrative review follows the guidance [22] and synthesizes the existing literature on how clinical educators influence the development of both clinical competence and psychological resilience in undergraduate nursing students during clinical placements. A narrative review is appropriate for this topic, because it offers an expert’s perspective on a topic relevant to clinical practice [23], and it allows for the integration of diverse study types, theoretical frameworks, and conceptual insights that inform complex and context-specific educational processes.

### 2.2. Search Strategy

To identify the relevant literature, a comprehensive search was conducted across three major databases: PubMed, CINAHL, and PsycINFO. These databases were selected due to their relevance to nursing, health sciences, and psychological research. The search strategy combined four key concept areas using Boolean operators: (1) nursing students, (2) clinical education settings, (3) teaching strategies and educator behaviors, and (4) outcomes such as resilience and competence. The following example search string was used in PsycINFO (Ovid): (“nursing students” or “undergraduate nursing” or “student nurse”).mp. [mp = title, abstract, heading word, table of contents, key concepts, original title, tests & measures, mesh word] AND (“clinical education” or “clinical placement” or “nursing education”).mp. [mp = title, abstract, heading word, table of contents, key concepts, original title, tests & measures, mesh word] AND (“teaching strategies” or “educator behaviors” or “instructional methods” or “clinical supervision”).mp. [mp = title, abstract, heading word, table of contents, key concepts, original title, tests & measures, mesh word] AND (“resilience” or “psychological resilience” or “clinical competence” or “emotional safety” or “self-efficacy” or “emotional intelligence”).mp. [mp = title, abstract, heading word, table of contents, key concepts, original title, tests & measures, mesh word]. Searches were limited to peer-reviewed articles published in English. Additional studies were identified through manual screening of reference lists from relevant articles to ensure a comprehensive inclusion of both theoretical and empirical works aligned with the review’s aim.

### 2.3. Inclusion and Exclusion Criteria

Studies were included in the review if they focused on nursing faculty or undergraduate nursing students engaged in clinical skills training (e.g., simulation), clinical education, or clinical placements, and explored teaching strategies, educator behaviors, or characteristics of the clinical learning environment. To be eligible, studies had to report outcomes related to clinical competence and resilience. Qualitative, quantitative, and mixed-methods studies published in peer-reviewed articles published between 1 January 2000 and 2 May 2025 in peer-reviewed journals were included. The specific time frame was selected to capture major shifts in nursing education over the past 25 years, including the integration of simulation, evolving clinical teaching strategies, and increased focus on competence in undergraduate clinical training.

Studies focused on postgraduate students, licensed professionals, or students from non-nursing disciplines were excluded. Articles that addressed general academic performance without reference to clinical competence or resilience, or that focused solely on educational technologies or online simulations without a clinical educator component, were excluded. Additionally, editorials, commentaries, conference abstracts, non-peer-reviewed publications, articles not available in full text, or not written in English were also excluded.

### 2.4. Study Selection

Titles and abstracts were screened for relevance by one reviewer (SK), followed by full-text screening of eligible articles (SK). The extracted data and disagreements were discussed and resolved through consensus within the review team. The final selection was based on the article’s contribution to at least one of the review’s core themes: clinical teaching, student competence, resilience, or the clinical learning environment. A PRISMA flowchart for the literature selection process is provided as a Appendix A.

### 2.5. Data Extraction and Synthesis

Key information relevant to the research questions was extracted from the selected studies and included: author(s), year, and country, study design and methodology, population and setting, teaching strategies or educator behaviors, and outcomes related to competence and/or resilience. The findings were synthesized thematically, guided by the conceptual framework based on Bandura’s SCT. Studies were grouped into the three core domains of the framework: teaching strategies and educator behaviors, characteristics of the clinical environment, and mediating factors. Themes and subthemes were identified deductively and organized to reflect how different educational strategies foster competence and resilience in students.

### 2.6. Quality Appraisal

As this is a narrative review, a structured quality appraisal of the included studies was not required [24]. Narrative reviews aim to provide a broad and integrative synthesis of literature across diverse study designs, theoretical approaches, and contexts.

## 3. Results

### 3.1. Characteristics of the Studies

This review included 19 studies published between 2016 and 2025, conducted across ten countries: the USA [25,26,27,28], England [29], Taiwan [30,31], Hong Kong [10], South Africa [32,33], Iran [34,35], Ireland [36,37], South Korea [38,39], the Philippines [40], and China [41,42]. Study designs included mixed-methods [25,36], quantitative quasi-experimental [26,34,35,39], cross-sectional and correlational [30,38,40,41,42], descriptive case study [27], qualitative focus groups [10], exploratory/descriptive qualitative studies [28,29,32,33,37], and participatory action research [31]. Populations included undergraduate or BSN students [10,25,26,29,30,31,32,33,34,35,36,38,39,40,41,42], mental health nursing students [37], and nursing faculty [28]. Studies examined both competence and resilience [10,25,26,27,28,31,32,33,34,35,36,37,38,40,41], while few studies focused on resilience [29,39] or competence [42]. Competence outcomes included clinical judgment, decision-making, emotional intelligence, work readiness, and clinical adjustment; resilience outcomes covered self-efficacy, stress management, ego-resiliency, adaptive learning, and psychological well-being. The majority employed descriptive, cross-sectional, or quasi-experimental designs, and many relied on self-reported outcomes from students or faculty. Sample sizes varied considerably, but several studies involved relatively small cohorts, especially in qualitative or mixed-methods designs. Few studies used validated clinical assessments (Table 1).

The findings showed that mindfulness, stress management, and self-care interventions strengthened resilience [25,29], while instructor and faculty support improved resilience, self-efficacy, and clinical adjustment [30,40]. Emotional intelligence training enhanced self-awareness, communication, and coping [28], and reflective practices and simulation-based learning improved competence, critical thinking, and resilience [31,36,37]. Quasi-experimental studies demonstrated gains in decision-making, self-efficacy, clinical communication, and psychological well-being [26,34,35,39]. Social support and learning environments positively influenced resilience and competence [33,41,42], while ego-resiliency mitigated the negative effects of social anxiety [38]. Higher resilience was linked to better clinical integration, self-awareness, satisfaction, and retention, particularly during the challenges associated with the COVID-19 pandemic [10,27] (Table 1).

### 3.2. Characteristics of Learning Environments

The included studies reported a wide range of learning environments shaping nursing students’ experiences. Supportive clinical or academic settings were common [28,30,33,38,40,41,42], along with safe and emotionally supportive environments [25,29,31,39]. Simulation-based learning environments with advanced setups were also frequently described [26,35,36], alongside inclusive and dialog-driven settings [27,37], structured bedside or clinical supervision [34,42], and socially supportive ward cultures [10]. However, challenging or hostile environments were also reported [32], highlighting important contextual variations in clinical learning environments (Table 2).

### 3.3. Teaching Strategies

#### 3.3.1. Simulation-Based Learning and Reflection

Several studies emphasized simulation as a foundational strategy, often paired with structured debriefing and reflection. Structured reflection and individualized resilience plans showed positive effects on student preparedness and confidence [25]. When paired with structured debriefing and guided reflection, simulation created safe spaces for students to practice technical skills, process complex scenarios, and build confidence in their abilities. Across the studies, these strategies were shown to enhance students’ preparedness, promote emotional regulation, and foster adaptive responses to clinical stress [10,25,36]. Whether delivered through web-based scenarios or high- and low-fidelity simulations, the inclusion of role play, mentorship, and extended debriefing allowed learners to engage deeply with both the clinical and emotional dimensions of care [26,30]. Repeated exposure through mastery learning further strengthened self-efficacy and reinforced clinical reasoning, highlighting the importance of structured, reflective practice as part of simulation-based education [35] (Table 2).

#### 3.3.2. Emotional and Psychological Skill Building

Several strategies specifically targeted emotional resilience and stress regulation. Life skills training was reported to improve students’ capacity to manage uncertainty and adapt to stress [37]. Poetry therapy provided students with a creative outlet for emotional processing and self-expression [39]. Mindfulness and reflective practices were used to promote emotional regulation and enhance resilience [29]. Psychological support interventions helped to buffer students from emotional exhaustion and supported mental well-being during clinical placements [38] (Table 2).

#### 3.3.3. Communication and Reflective Practice

Teaching strategies also focused on improving communication and fostering reflective practice. ISBAR-based supervision improved students’ ability to communicate clearly and confidently in clinical settings [34]. Formal teaching on communication and reflective practice contributed to students’ self-awareness and interpersonal competence [28]. Reflective workshops created space for students to explore their experiences, process emotions, and develop critical thinking [31] (Table 2).

#### 3.3.4. Experiential and Inclusive Learning

Several studies emphasized experiential, student-centered, and inclusive strategies that promoted both competence and resilience. Experiential learning approaches gave students opportunities to engage with complex patient scenarios and develop problem-solving skills [32]. Inclusive pedagogy fostered equitable learning environments that accommodated diverse needs [27]. Structured mentorship and feedback allowed for personalized learning and adaptive support [40]. The establishment of social support networks strengthened students’ sense of belonging and promoted emotional safety [41]. While supportive clinical environments that emphasized psychological safety and active supervision linking to better learning outcomes [42], mentorship promoted student independence and helped build confidence and transition readiness [33] (Table 2).

### 3.4. Educator Behaviors

In addition to strategies, educator behaviors directly influenced how students experienced clinical education.

#### 3.4.1. Feedback, Guidance, and Supervision

Effective educator behaviors were essential in promoting both resilience and clinical competence. Structured feedback and guidance allowed students to reflect on their strengths and areas for growth [25]. Supportive clinical supervision, including continuous presence and timely feedback, enhanced students’ confidence and reduced their stress [30,42]. Simulation facilitation guided by experienced educators enabled safe skill development and reflective learning [26]. Bedside supervision contributed to real-time learning and emotional reassurance [34]. Formative feedback, when consistently delivered, strengthened students’ sense of self-efficacy and improved clinical judgment [35] (Table 2).

#### 3.4.2. Facilitating Reflection and Peer Learning

Educators played a central role in encouraging reflection and peer interaction. Facilitating reflection and fostering social support networks helped students make sense of emotionally complex situations [10]. Educators who encouraged peer learning fostered collaboration, mutual support, and team-based problem-solving [36]. Reflective mentorship helped students develop self-awareness and resilience in the face of clinical challenges [31] (Table 2).

#### 3.4.3. Psychosocial Support and Role Modeling

Educators served as both emotional anchors and role models, shaping students’ resilience and professional identity. Proactive emotional support helped students manage stress and maintain motivation during clinical placements [27], while psychological support promoted emotional regulation and coping with adversity [38]. Supportive mentorship fostered safe, growth-oriented environments and strengthened students’ professional identity [40]. By encouraging student-led support networks, educators also promoted connection and resilience [41]. Inclusive and compassionate role modeling contributed to psychologically safe learning spaces where students felt respected and engaged [28,29]. Therapeutic approaches, including life skills training and poetry therapy, supported holistic development [37,39]. Mentorship that emphasized autonomy helped prepare students for independent practice and professional confidence [33] (Table 2).

### 3.5. Implications for Practice

A consistent recommendation across studies was the integration of resilience-building and reflective practices into the curriculum, highlighting their role in helping students manage stress and develop professional identity [10,25,28,29,36,37,38,39,40]. Structured simulation and clinical supervision were frequently emphasized as effective methods for developing competence and self-efficacy [26,34,36]. Supportive learning environments characterized by mentorship, regular feedback, and psychological safety, were also highlighted as essential for student growth and engagement [27,30,31,32,33,41,42]. Several studies emphasized the need to better align theory with clinical practice through structured support systems and reflective learning opportunities [10,31,32]. In addition, innovations in curriculum design were proposed, including the integration of emotional intelligence training [28], mastery learning approaches [35], and creative modalities such as art-based therapy [39] (Table 2).

### 3.6. Mediating Factors

Several mediating factors were found to promote resilience and competence in nursing students. Supportive relationships and instructor support were consistently emphasized [27,28,30,31,32,33,34,39,41,42]. Reflective practices and structured debriefings were widely used to enhance self-awareness and coping [10,25,26,28,29,31,36,37,39]. Peer support and engagement further contributed to resilience [27,31,36]. Feedback mechanisms, including frequent, individualized, and formative feedback, also supported learning and confidence [33,34,35]. Conversely, poor interpersonal support and ineffective communication were associated with negative outcomes [32] (Table 3).

### 3.7. Influence on Competence and Resilience

The studies reported multiple mediating factors influencing competence and resilience. Support and reflection improved student commitment to self-care, coping, and adaptability [25,28,33] and enhanced emotional regulation and interpersonal skills [29,37]. Instructor support and feedback improved self-efficacy and competence [30,34,35], while peer learning and reflective practices strengthened clinical judgment and resilience [31,36]. Technical preparation and structured debriefing led to higher competence, confidence, and knowledge retention [26]. Psychological resilience factors mediated stress and improved competence [10,38,39,40]. Finally, work readiness and clinical competence were enhanced through social support and self-regulated learning [41,42] (Table 3).

## 4. Discussion

This review provides a comprehensive synthesis of how teaching strategies, educator behaviors, and clinical environments foster both clinical competence and psychological resilience in nursing students, a combination that is underexplored in prior reviews. Using Bandura’s SCT as a framework, this review offers a unique perspective that bridges the gap between technical skill acquisition and emotional adaptability in clinical education.

### 4.1. Teaching Strategies and Educator Behaviors

This review highlights that clinical educators play a foundational role in shaping nursing students’ development of both clinical competence and psychological resilience. Consistent with SCT, students learn not only through formal instruction, but also through observational learning, where educator behaviors serve as powerful models for clinical conduct, emotional regulation, and communication [14,43]. Educators who consistently demonstrated compassionate care, reflective thinking, and ethical decision-making provided students with concrete examples of how to act under pressure [28,29]. These modeling experiences are central to the development of self-efficacy, which is a core SCT construct linked to motivation and performance [44].

However, our review reveals that students often experience fragmented role modeling due to the short duration of clinical rotations and lack of continuity with assigned mentors. This disruption limits opportunities for pattern recognition and relational learning, which are vital for professional identity formation. Adding to the challenge, clinical educators frequently report insufficient time, inadequate training, and a lack of institutional support for their teaching responsibilities [45,46]. Without consistent and engaged role models, students may struggle to internalize professional behaviors, which can undermine their confidence and clinical judgment. Addressing these gaps requires deliberate faculty development and better alignment between academic institutions and clinical sites. Structured mentorship programs, protected teaching time, and pedagogical training for clinical staff are essential to ensure conditions where role modeling can consistently occur and be recognized as a strategic teaching act.

#### 4.1.1. Mastery Learning Through Simulation and Feedback

Simulation-based education, especially when combined with structured debriefing and reflective dialog, emerged as a highly effective approach for promoting self-efficacy and adaptive coping. These structured learning experiences allow students to engage in mastery learning, which Bandura identifies as the most potent source of efficacy beliefs [44]. Simulation offers a psychologically safe environment where students can practice clinical decision-making, make mistakes without judgment, and engage in real-time self-assessment [25,26]. As previous studies have shown, simulation enhances confidence, improves clinical skills, and reduces stress by providing emotional distance from high-pressure clinical encounters [47,48]. Despite these benefits, implementing simulation and structured feedback in low-resource settings can be challenging due to the cost, limited staffing, and infrastructure constraints. Moreover, the effectiveness of the simulation may vary depending on students’ baseline confidence, prior clinical exposure, and emotional regulation skills. Students with high anxiety may perceive simulated failure as confirmation of inadequacy, weakening rather than enhancing self-efficacy. This illustrates SCT’s emphasis on the moderating role of affective states in learning outcomes. However, scalable adaptations such as low-fidelity simulations using basic mannequins, role-play, or scenario-based discussions offer viable, cost-effective alternatives for skill-building in low-resource environments [42]. These methods preserve the core SCT principle of mastery learning by enabling repeated, guided practice, even when high-tech resources are unavailable. Addressing contextual constraints is essential to ensure equitable access to resilience-building interventions across diverse clinical education systems.

Feedback itself plays a central role in developing self-efficacy. Verbal encouragement, formative assessments, and constructive critique function as “verbal persuasion,” a recognized source of efficacy development in SCT [30,34]. When feedback was timely, specific, and affirming, students gained clarity on their clinical strengths and areas for growth, reinforcing their confidence to perform under pressure. Nevertheless, access to high-quality simulation and educator feedback remains uneven across settings. Institutional investment in simulation labs and formal feedback training should be prioritized to ensure equitable access to these evidence-based strategies. From an SCT perspective, simulation-based learning supports two of the most powerful sources of self-efficacy: mastery experiences and verbal persuasion. Mastery experiences are achieved through repeated, hands-on practice in safe environments, while structured debriefings and instructor feedback offer verbal reinforcement that boosts confidence. This dual mechanism may explain why simulation strategies are particularly effective in high-resource settings, where these components are reliably integrated into the curriculum. Moreover, simulation may unintentionally elevate physiological arousal, such as stress or embarrassment, particularly for students unfamiliar with high-stakes assessments. According to SCT, such emotional responses may negatively influence efficacy beliefs. This highlights the need for educators to monitor not only technical outcomes, but also affective responses during simulation to ensure positive efficacy reinforcement.

#### 4.1.2. Social Learning and Peer-Based Resilience

Social learning also plays a critical role in fostering resilience. In line with SCT’s emphasis on social reinforcement, peer mentoring, group reflection, and interpersonal dialog contributed to emotional regulation, a sense of belonging, and reduced academic stress [31,36]. Supportive peer networks helped normalize students’ emotional experiences, enabling them to cope more effectively with clinical adversity. These findings align with existing research showing that peer-based learning environments enhance psychological safety and mitigate burnout risks [10,49,50]. While peer-based strategies are broadly effective, their impact may vary across educational cultures and settings. For instance, in collectivist learning environments, students appeared to benefit strongly from vicarious learning through peer modeling and group reflection. In contrast, studies from more individualistic or high-autonomy systems, such as the United States and Ireland, emphasized the value of direct feedback and structured individual reflection. These patterns suggest that the social learning mechanisms emphasized in SCT may be differentially activated depending on cultural norms around collaboration, hierarchy, and independence. Additionally, within the same cultural context, students may differ in how they respond to peer feedback based on personality traits (e.g., introversion), perceived peer competence, or academic standing. SCT’s focus on personal factors within triadic reciprocal determinism supports the idea that individual variation shapes how observational learning and verbal persuasion are internalized.

Experiential and creative methods such as role play, mindfulness, poetry therapy, and resilience planning also demonstrated utility in strengthening students’ emotional flexibility and adaptive coping. These approaches enabled students to process ambiguity, manage uncertainty, and engage in self-regulation, reinforcing the idea that resilience is not innate but can be cultivated through intentional educational design [16,51]. However, peer-based learning benefits are not automatic. Barriers such as unequal access to peer support, differences in student openness to feedback, and lack of structured reflection opportunities can limit their effectiveness. Faculty can address these challenges by intentionally scaffolding peer interactions through group assignments, co-reflection exercises, and routine peer debriefing sessions.

The review also highlights that the same educator strategies can yield different outcomes depending on the context in which they are applied. For instance, frequent feedback may have a stronger impact in small-group or high-contact learning environments, where students are more likely to trust and internalize educator input. Conversely, in resource-limited or high-stress placements, physiological arousal such as anxiety or emotional overload may undermine self-efficacy, even when support mechanisms are present. SCT recognizes these factors as key moderators of learning and adaptation, reinforcing the importance of tailoring interventions to the emotional and logistical realities of different clinical settings.

While peer strategies are broadly effective, their impact may depend on students’ initial self-efficacy beliefs and personal coping mechanisms. For instance, students with high academic anxiety or limited clinical exposure may interpret peer success as threatening rather than motivating, especially in the absence of emotional support. SCT emphasizes that such negative vicarious experiences, if unbuffered, can reduce rather than enhance self-efficacy. This suggests the importance of intentionally scaffolding peer-based learning for vulnerable student subgroups through co-reflection or instructor-facilitated group processing.

### 4.2. Clinical Learning Environment

The structure and culture of the clinical learning environment had a significant impact on how students developed both competence and resilience. Environments characterized by psychological safety, consistent supervision, inclusive communication, and structured learning activities allowed students to engage confidently and meaningfully with clinical care [52]. In such settings, students reported feeling respected, valued, and more integrated into the professional team. In contrast, variability in the quality of placements led to stark differences in student experience. Some students encountered emotionally unsafe or disorganized settings that undermined their learning and well-being. These negative experiences align with SCT’s identification of “physiological and affective states” as key sources of self-efficacy. Persistent stress, lack of inclusion, or emotional exhaustion can serve as negative cues, weakening students’ belief in their ability to succeed and ultimately reducing both resilience and clinical performance. This underscores SCT’s recognition that self-efficacy is not built uniformly, and external stressors and negative emotional cues can override the benefits of otherwise strong educator input. In emotionally unsafe environments, even excellent role modeling may fail to influence behavior due to disrupted cognitive processing and reduced social trust.

These findings aligned with the findings of previous research on the inconsistent quality of clinical supervision across placement sites [53]. Such disparities underscore the need for standardized preceptor training and institutional mechanisms for monitoring the quality of clinical education. From the perspective of SCT, the learning environment represents the critical third element of Bandura’s triadic model (personal, behavioral, and environmental factors). Without a psychological and pedagogically sound context, neither observational learning nor self-efficacy can be reliably developed. Institutions must therefore view the clinical setting not just as a backdrop for practice, but as an active driver of student outcomes.

Furthermore, the clinical environment operates in reciprocal interaction with student beliefs and behaviors, consistent with SCT’s triadic model of personal, behavioral, and environmental influences. Supportive placement may enhance learning outcomes for students with high self-efficacy, while those with prior negative experiences or emotional distress may require more explicit modeling and encouragement to engage confidently. Recognizing this bidirectional dynamic can inform the design of adaptive supervision strategies tailored to individual learners’ needs.

### 4.3. Implications for Nursing Education

Our findings support a shift from traditional competency-based models toward a holistic, resilience-informed framework for clinical education. While clinical competence ensures technical proficiency, resilience safeguards long-term emotional sustainability which are essential in the context of increasingly complex and demanding healthcare systems. To prepare for a future nursing workforce that is both clinically skilled and psychologically resilient, institutions must treat resilience not as a soft skill, but as a core educational outcome. Embedding SCT-informed strategies such as role modeling, simulation, guided reflection, and peer support into national nursing curricula, backed by policy-level investment in faculty development and placement quality assurance, is essential. These measures can enhance student learning, reduce attrition, and contribute to a more sustainable and adaptable health workforce. To enhance the effectiveness of resilience-building strategies, educational institutions should tailor interventions to the self-efficacy mechanisms most applicable within their specific context. In environments where educator availability is limited, approaches that leverage peer mentorship and observational learning may be more practical than those requiring intensive one-on-one feedback. However, institutions should also consider student diversity in cognitive and emotional readiness. For example, first-year students or those from underrepresented backgrounds may require more structured feedback to develop confidence, whereas more experienced students may benefit from autonomy and peer co-reflection. This highlights SCT’s principle that learning outcomes emerge from the dynamic interplay of personal, behavioral, and environmental factors. Additionally, cultural adaptation may be necessary to ensure relevance and impact, particularly regarding norms around emotional expression, authority dynamics, and student–teacher interaction.

### 4.4. Limitations

This review has several limitations that should be acknowledged. First, although a comprehensive literature search was conducted, inclusion was limited to peer-reviewed studies published in English between January 2000 and May 2025. This language restriction introduces potential selection bias and may exclude relevant studies from non-English-speaking regions, particularly those in low- and middle-income countries. As a result, the findings may disproportionately reflect perspectives from high-income settings with well-resourced clinical education systems. Second, no formal critical appraisal of study quality was conducted. While this aligns with narrative review methodology, it limits our ability to assess the rigor, credibility, and risk of bias in the included studies. Many relied on small samples, descriptive designs, or self-reported outcomes, which may affect the strength and transferability of the evidence. Future reviews may benefit from applying standardized tools such as the JBI critical appraisal checklists or the Mixed Methods Appraisal Tool (MMAT) to enhance methodological transparency. Third, most included studies were conducted in high-resource environments. The under-representation of low-resource contexts limits the generalizability of our findings, particularly in settings where access to simulation, structured supervision, or faculty development is constrained. Fourth, while Bandura’s SCT provided a valuable framework for synthesis, this deductive approach may have limited the emergence of alternative models or unanticipated insights. Broader conceptual frameworks and participatory research approaches may yield a more nuanced understanding in future work. Fifth, neither students nor clinical educators were involved in the design, selection, or interpretation stages of this review. Incorporating their perspectives could have strengthened the contextual relevance, especially regarding the lived realities of clinical education. Finally, most studies reported short-term outcomes. There was limited evidence on the long-term impact or sustainability of interventions aimed at building resilience and competence, which highlights the need for more longitudinal research in this area.

## 5. Conclusions

This narrative review highlights that clinical competence and psychological resilience are co-constructed outcomes in nursing education, shaped through educator behaviors, targeted teaching strategies, and supportive clinical learning environments. By applying Bandura’s SCT, the review synthesizes how self-efficacy, observational learning, and emotional support enable students to develop both professional capabilities and adaptive strengths. This study presents one of the first theory-informed syntheses that link resilience and competence development through a unified educational framework, addressing a gap in the fragmented literature. The evidence supports a shift away from isolated or informal practices toward structured, resilience-informed educational design. Integrating simulation-based learning with guided debriefing, consistent feedback, reflective practice, and peer-supported learning into nursing curricula can help students manage the emotional demands of clinical work while strengthening clinical reasoning and confidence. These elements are not supplementary but foundational to effective and sustainable clinical education. For institutions, the findings point to clear and actionable priorities. Investing in educator training, protecting time for clinical teaching, and enhancing placement quality are crucial to enable educators to serve as effective role models and facilitators. Aligning expectations between academic and clinical environments and developing joint supervision models or shared educator roles may further improve continuity and student learning outcomes. Embedding these strategies into national curricula and accreditation frameworks could also promote more consistent and equitable educational quality.

Future research should extend beyond short-term evaluations to investigate the long-term effects of resilience-building strategies. Longitudinal, multicenter studies that reflect diverse cultural and resource contexts are especially needed. Involving students and educators in the co-design and evaluation of interventions will enhance their relevance, feasibility, and impact. Supporting resilience not only strengthens clinical competence, but also contributes to student well-being, reduces attrition, and fosters a more emotionally prepared and sustainable nursing workforce. This review provides educators, researchers, and institutional leaders with practical, evidence-informed directions to inform educational planning, policy, and implementation.

## Figures and Tables

**Table 1 nursrep-15-00253-t001:** Summary of the included studies and key results.

Author(s)/ Year/Country	Study Design	Population/Setting	Aim of Study	Type of Outcome	Measurement	Key Results
Carter et al., 2023, USA [25]	Mixed-methods design	31 sophomore nursing students in a Bachelor of Science in Nursing (BSN) program, Midwestern US university	To explore preferred resilience-building strategies and their impact on future nursing practice through the Student Nurse Resiliency Project (SNRP)	Resilience	Student self-report via structured reflection papers, frequency of strategy use	Students preferred mindfulness, stress management, and exercise; strategies were found effective in promoting resilience Mindfulness, stress management, and self-care were most effective; resilience linked to better coping, mental health, and care quality
Chen et al., 2021, Taiwan [30]	Quantitative	101 senior nursing students in clinical practice	Examine effects of web-based interactive situational teaching vs. traditional teaching on clinical performance and self-efficacy	Competence (clinical judgment) Resilience (self-efficacy)	Clinical judgment via LCJR, self-efficacy via GPSE scale	The degree of instructor support during the internship process significantly improved student resilience self-efficacy in clinical performance Traditional teaching showed better clinical competence outcomes
Ching et al., 2020, Hong Kong [10]	Qualitative—focus group interviews	24 final-year baccalaureate nursing students in Hong Kong hospitals	Explore stressors and coping strategies of nursing students with differing levels of resilience and burnout during clinical placements	Resilience Clinical competence	Qualitative thematic analysis, focus groups, resilience scores (CD-RISC) Clinical performance and adaptability	High resilience linked to self-awareness Low resilience linked to self-blame and external coping strategies Self-directed students achieved better clinical integration
Curl et al., 2016, USA [26]	Quantitative—quasi-experimental	124 associate degree nursing students across three nursing schools, clinical specialties (OB, pediatrics, mental health, critical care)	Evaluate the effectiveness of replacing clinical experiences with simulation	Competence Resilience	HESI medical–surgical specialty exams, (OB, pediatrics, mental health) National Council Licensure Examination (NCLEX) pass rates Student evaluations (Likert-scale surveys)	Improved nursing knowledge, and clinical competence measured by HESI exams. Improved critical thinking and confidence Debriefing is critical for translating simulation experiences into clinical readiness
Dowling et al., 2021, USA [27]	Quantitative—descriptive case-study	Undergraduate nursing students at University of Virginia School of Nursing	To describe how nursing programs create resilient, inclusive learning environments fostering student competence and resilience during COVID-19	Competence Resilience	Course evaluations, retention/graduation rates, anecdotal student/faculty feedback, NCLEX pass rates	High satisfaction, improved belongingness, steady retention rates, increased resilience during the COVID-19 pandemic
Fadana & Vember, 2021, South Africa [32]	Qualitative	38 undergraduate nursing students, healthcare facilities, Boland Overberg, Western Cape	To explore and describe the experiences of undergraduate student nurses during clinical practice in healthcare facilities	Competence Resilience	Objective structured clinical examination (OSCE) performance, student reports, Qualitative interviews	Negative experiences hindered competence development (theory-practice gap, anxiety, lack of supervision) Anxiety and decreased resilience (stress, discrimination)
Gheisari et al. (2025), Iran [34]	Quantitative—quasi-experimental (pre–post, two groups)	80 nursing internship students, medical and surgical wards of the Isfahan University of Medical Sciences	Investigating the impact of ISBAR-based clinical supervision model on clinical decision-making and self-efficacy	Competence (clinical decision-making) Resilience (self-efficacy)	Clinical decision-making questionnaire (24-item Likert scale) Self-efficacy in clinical performance questionnaire (37-item Likert scale)	Improved clinical decision-making, increased clinical self-efficacy, improved handover communication skills after ISBAR-based clinical supervision
Hill et al., 2023, Ireland [36]	Mixed-methods	17 4th-year BSc nursing students; University College Dublin, Ireland	Evaluate effectiveness of low and high-fidelity simulations for internship preparation	Competence Resilience	Self-reported student surveys (knowledge, skills, decision-making, confidence) Qualitative feedback and open-ended comments (reflection, coping, confidence, anxiety)	Improvements in clinical competence (skills, critical thinking, decision-making, knowledge retention) Increased confidence in clinical scenarios, reduced anxiety Improved resilience, reduced anxiety, enhanced coping through reflective practice during simulations
Ireland, 2022, USA [28]	Qualitative—descriptive	8 full-time didactic nursing faculty/8 U.S. public universities	Explore how faculty incorporate emotional intelligence (EI) competencies into baccalaureate nursing education	Competence (EI skills) Resilience (stress management)	Faculty-reported integration in curricula Reflective practice and feedback	EI improved self-awareness, communication, empathy, interpersonal skills EI fostered student preparedness and students better managed workplace stress and incivility
Jun & Lee, 2017, South Korea [38]	Quantitative—cross-sectional survey	329 nursing students/three nursing colleges in South Korea	To identify the role of ego-resiliency in the relationship between social anxiety and problem-solving ability in nursing students	Competence (problem-solving) Resilience (ego-resiliency)	Validated self-reported Social Problem-Solving Inventory (SPSI) scale	Ego-resiliency partially mediated the negative relationship between social anxiety and problem-solving ability Social anxiety negatively correlated with competence; ego-resiliency positively correlated and mediated the relationship
Labrague et al., 2025, Philippines [40]	Quantitative—online survey	506 undergraduate nursing students from three nursing schools in the Philippines	To examine the role of psychological resilience as a mediator between nurse faculty support and clinical adjustment	Competence (clinical adjustment) Resilience	Brief Resilience Scale (BRS), Clinical Adjustment Scale (CAS-SN)	High faculty support associated with higher resilience and better clinical adjustment
Liang et al., 2019, Taiwan [31]	Qualitative—participatory action research	28 senior nursing students in Taiwanese hospital-based practicum	To develop and implement a resilience enhancement (RE) project during clinical practice	Competence Resilience	Qualitative content analysis from group discussions, interviews, reflective diaries Observations of clinical performance and peer feedback	Improved clinical competence (skills, communication, and knowledge integration) Enhanced self-exploration, confidence, and emotional resilience through RE workshops
Ma et al., 2024, China [41]	Quantitative—cross-sectional	376 final-year nursing students in China	To examine the relationships among social support, resilience, and work readiness	Competence (work readiness) Psychological resilience	Work Readiness Scale for Graduate Nurses (WRS-GN), Connor–Davidson Resilience Scale (CD-RISC)	Perceived social support improves resilience, which in turn improves work readiness Resilience partially mediated social support’s effect
Mehdipour-Rabori et al., 2021, Iran [35]	Quantitative—quasi-experimental (pre-test–post-test, two-group)	105 bachelor nursing students, Kerman University of Medical Sciences	To assess the effect of simulation-based mastery learning on clinical skills	Competence (clinical skills) Resilience (adaptive learning)	Checklist scores (0–144), clinical skill exams Qualitative feedback on iterative practice	Significant improvement in clinical skills scores (e.g., suction, nasogastric tube feeding) in intervention group
O’Sullivan et al., 2021, Ireland [37]	Qualitative—descriptive	10 mental health student nurses in clinical placement	Explore students’ perceptions and application of the Decider Life Skills program	Competence Resilience	Focus group interviews post-placement	Improved confidence, coping skills, stress management, and resilience in clinical settings
Park et al., 2022, South Korea [39]	Quantitative—quasi-experimental, non-equivalent control group, pretest–posttest	49 senior nursing students from two colleges in Jeonju, South Korea	To examine the effects of a poetry therapy program on stress, anxiety, ego-resilience, and psychological well-being	Resilience (ego-resilience, psychological well-being)	Pre-/post-follow-up surveys (validated scales for stress, anxiety, resilience)	Significant improvements in ego-resilience and psychological well-being and reduction in stress/anxiety
Stacey et al., 2017, England [29]	Qualitative—exploratory case study	21 nursing students, 5 facilitators; University of Nottingham	To evaluate resilience-based clinical supervision (RBCS) in developing resilience-based competencies	Resilience	Focus groups pre-, post-, and 6 months after intervention	Improved self-care (mindfulness, boundary setting), distress tolerance, empathy, and compassionate values
Yu et al., 2021, China [42]	Quantitative—cross-sectional	1518 undergraduate nursing students/five medical colleges in China	Examine clinical competence and its association with self-efficacy and clinical learning environments	Clinical competence	Holistic Clinical Assessment Tool (HCAT-C), General Self-Efficacy Scale (GSES), Clinical Learning Environment, Supervision and Nurse Teacher Evaluation Scale (CLES+T)	Clinical competence positively associated with self-efficacy and clinical learning environment Students in supportive environments showed higher competence
Zulu et al., 2021, South Africa [33]	Qualitative—descriptive	25 fourth-year nursing students/primary healthcare clinics (PHC) clinics in one South African province	To explore and describe nursing students’ experiences and support during clinical placement in PHC settings	Competence Resilience	Thematic analysis of focus group interviews Student narratives on coping strategies	Students developed clinical skills, cultural competence, confidence, and resilience despite challenges Supportive supervision and intrinsic motivation build resilience; real patient care builds competence

**Table 2 nursrep-15-00253-t002:** Characteristics of learning environments supporting student competence and resilience.

Author(s), Year, Country	Learning Environment	Teaching Strategy	Educator’s Behavior	Results on Competence, Resilience	Impact on Students	Implication for Practice
Carter et al., 2023, USA [25]	Supportive faculty feedback, evidence-based assignment structure, emotional safety	Competency-based, active-learning via structured writing and reflection, individualized resilience plans	Feedback on paper structure, guidance through structured reflection and strategy implementation	Increased awareness and application of self-care strategies; enhanced resilience	Improved mental health, academic performance, and perceived preparedness for clinical stressors	Embed resilience training early in curricula to foster lifelong coping and reflective skills
Chen et al., 2021, Taiwan [30]	Digital learning environment with face-to-face discussion Supportive instructor behaviors during clinical supervision	Web-based interactive situational teaching (scenario-based exercises, online ethical decision-making model) vs. traditional teaching	Supportive instructor during clinical practice	Traditional teaching was more effective in enhancing clinical judgment and internship scores Higher self-efficacy in students with strong instructor support	Increased clinical judgment skills and improved self-efficacy due to instructor support	Combine digital tools with real-world application tasks Train preceptors in supportive feedback and mentorship
Ching et al., 2020, Hong Kong [10]	Supportive, socially inclusive ward culture with clear communication Busy clinical wards with high workload	Structured debriefing sessions, mentorship programs	Facilitating self-awareness, reflective encouragement, providing feedback, fostering social support	Improved self-awareness in high-resilience students, external coping in low-resilience students Self-directed students showed higher resilience and lower burnout	Enhanced professional identity and coping skills Reduced burnout among reflective, self-directed students	Clinical educators should foster autonomy, self-awareness, and active fitting into clinical environments Incorporate reflective practices and resilience training in clinical education
Curl et al., 2016, USA [26]	Centralized simulation lab with scenarios mirroring clinical specialties (OB, pediatrics, mental health, critical care)	Simulations and clinical experiences (STRIPES) intervention: 20 simulation modules (5 per specialty) with 1) pre-lab preparation, 2) active participation in high-fidelity simulation, and 3) extended debriefing (45–90 min)	Minimal overt interaction, facilitated structured simulations, active student roles, comprehensive debriefing sessions	Higher HESI exam scores; improved self-reported confidence and critical thinking. No difference in clinical performance evaluations	Doubled clinical capacity by sharing placements. Students applied simulation skills in real clinical settings	Integrate structured simulations of clinical education if paired with pre-lab and debriefing Use simulations for high-risk/low-frequency scenarios
Dowling et al., 2021, USA [27]	Virtual/in-person hybrid cohort-based support Safe spaces for dialog Inclusive environment promoting belonging, diverse perspectives, and proactive support systems	Inclusive pedagogy, holistic admissions, peer tutoring, mentoring, resilience-oriented strategies	Faculty training in inclusivity, equity and anti-bias strategies; proactive student support; fostering community	Increased self-efficacy, belongingness, academic success, resilience, and reduced attrition	Students reported feeling respected, valued, and supported Increased student retention and satisfaction	Educational programs should intentionally implement structured inclusive teaching practices, equity training, and proactive student support
Fadana & Vember, 2021, South Africa [32]	Hostile, congested, discriminatory environment, poor interpersonal relationships	Experiential learning through clinical placements	Poor support, neglectful, hostile behaviors by clinical staff	Decreased competence (poor skill integration) Reduced resilience (increased anxiety and low confidence)	Negative emotional impact Demotivation, desire to quit	Need structured support, improved interpersonal relationships, align theory and practice, and smaller student groups Regular staff training to align practice with curriculum
Gheisari et al. (2025), Iran [34]	Structured, supportive, regular bedside supervision sessions	ISBAR-based clinical supervision model	Feedback provision, observational assessments, supportive interaction, regular bedside clinical supervision	Significant improvements in clinical decision-making and self-efficacy, enhanced clarity in handover communication	Increased confidence and competence in clinical tasks, better resilience through structured support	Implementation of ISBAR-based structured clinical supervision recommended for enhancing competence and self-efficacy (resilience) in nursing practice
Hill et al., 2023, Ireland [36]	Safe, controlled simulation labs with realistic clinical scenarios, pre-brief/debrief sessions	High- and low-fidelity simulations, pre-simulation videos, role-playing debrief sessions	Supportive, constructive feedback, encouragement of critical thinking, realistic role-play facilitation Encouraged reflection and peer learning	Increased clinical competence (critical thinking, procedural and decision-making skills), boosted confidence, resilience through reflective learning	Reduced anxiety about clinical practice; better preparedness for real-life clinical placements	Promote structured reflective practices and regular simulation experiences to build emotional resilience in clinical settings
Ireland, 2022, USA [28]	Supportive, inclusive, and experiential settings	Formal (communication, empathy, professional formation); Informal (experiential learning, role modeling, reflective practice)	Role modeling, inclusivity, team building, fostering discussion, experiential learning, formative feedback	Increased emotional intelligence (EI) competencies (self-regulation, empathy, social skills)	Better conflict management, reduced stress, improved teamwork	Integrate EI formally into curricula Embed resilience-building activities (e.g., debriefing)
Jun & Lee, 2017, South Korea [38]	Supportive clinical and academic environment that reduce stress	Psychological support strategies (e.g., reducing social anxiety, fostering ego-resiliency)	Suggested focus on psychological support and build ego-resiliency	Improved problem-solving ability with higher ego-resiliency	Better coping with stress and improved clinical decision-making skills	Integrate psychological resilience training and anxiety-reduction strategies into curricula
Labrague et al., 2025, Philippines [40]	Safe, inclusive, feedback-rich clinical settings	Supportive mentorship, (listening, encouragement), constructive feedback, role modeling guidance from nurse faculty	Supportive, approachable, inclusive, recognition of achievements	Higher faculty support increased resilience and clinical adjustment. Positive adjustment, increased resilience and competence	Better adaptation to clinical practice, lower stress	Train faculty in supportive behaviors and resilience-building techniques
Liang et al., 2019, Taiwan [31]	Emotionally safe, peer-supported, skill-enhancing, non-graded environment	Six RE workshops incorporating self-confidence, coping strategies, competency, positive thinking, peer discussion, mentor presentations, clinical skill-based practice Reflective diaries	Mentorship without grading, facilitating workshops, arranging lab practice	Increased self-efficacy, greater confidence, improved skills, enhanced self-reflection, and resilience	Reduced anxiety, stronger peer networks, and improved clinical adaptability	Use structured, supportive, reflective clinical education models to foster student development Use peer-led debriefing to bridge theory-practice gaps
Ma et al., 2024, China [41]	Supportive clinical practice environment (family, friends, mentors)	Social support networks	Implied encouragement of social support systems	Significant increase in work readiness through increased resilience	Better transition into workforce, reduced stress	Integrate structured social support (e.g., mentorship, peer networks) into clinical training
Mehdipour-Rabori et al., 2021, Iran [35]	Proficiency workshop with advanced moulage labs and structured checklists; individualized feedback	Simulation-based mastery learning with formative assessments, feedback, and repeated practice	Formative feedback, individual guidance, daily observation	Significant clinical skill improvement and reduced errors; increased engagement	Improved skill mastery; increased confidence, adaptability to clinical demands	Encourage implementation of mastery learning for skill development to boost resilience and competence Replace time-based training with competency-focused SBML in nursing curricula
O’Sullivan et al., 2021, Ireland [37]	Community and acute mental health clinical placements	One-day cognitive behavioral therapy/dialectical behavior therapy (CBT/DBT)-based decider life skills training (interactive, role-play, visual aids)	Use of role-play, interactive teaching, visual aids	Increased self-efficacy, coping skills, and emotional regulation	Greater confidence in clinical practice, emotional regulation, and readiness for group facilitation	Integrate resilience skill-building workshops into clinical nursing curricula
Park et al., 2022, South Korea [39]	Emotionally supportive, peer empathy, structured therapy space	Group poetry therapy sessions using ADDIE and Mazza’s resilience educational support (RES) model	Facilitation of group poetry reading, writing, sharing, and supportive dialog	Increased ego-resilience and well-being, reduced stress	Improved stress management and adaptability	Art-based therapy can support student mental health. Incorporating poetry therapy into nursing education can strengthen resilience
Stacey et al., 2017, England [29]	Emotionally safe, structured, trust-based supervision groups	Resilience-based clinical supervision (RBCS), mindfulness, reflective discussion, positive reframing	Modeling compassion, non-judgment, and structured reflection, providing a safe space, guided discussion	Improved coping skills, reflection, emotional regulation, and resilience	Increased self-care, reduced burnout, maintained compassionate practice despite workplace stressors	Integrate RBCS into nursing curricula Train facilitators in emotional regulation and reflective practice
Yu et al., 2021, China [42]	Supportive supervision, pedagogical atmosphere, structured clinical rotations, ethical role modeling	Clinical practice in supportive learning environments	Supportive supervision; culturally and ethically sensitive practice, feedback, role modeling	Higher self-efficacy and clinical competence in supportive environments	Greater competence, ethical awareness, professional development	Foster supportive clinical environments with mentorship and structured feedback
Zulu et al., 2021, South Africa [33]	Supportive, conducive atmosphere; resource-limited but opportunity-rich primary healthcare (PHC) clinics	Mentorship, independence-fostering, hands-on clinical exposure Clear communication between tutors and clinics	Welcoming attitude Encouraging independence Mentoring and guidance	Competence (skill application, autonomy) Resilience (persistence despite challenges)	Growth in responsibility, increased confidence, sense of belonging, independence, and professional identity	Use trained supervisors for student supervision, improve PHC placement support and feedback, improve tutor–clinic collaboration

**Table 3 nursrep-15-00253-t003:** Mediating educational factors and their influence on competence and resilience.

Author(s), Year, Country	Mediating Factor	Description	Theoretical Mechanism	Influence on Competence/Resilience
Carter et al., 2023, USA [25]	Support, engagement, reflection	Active participation in selecting and applying resilience strategies, guided reflection on their effects	Self-efficacy, reflective practice	Greater student commitment to self-care, better coping with academic and professional stress
Chen et al., 2021, Taiwan [30]	Instructor support Course applicability	Degree of perceived support provided by clinical instructors during clinical practice	Self-efficacy	Improved self-efficacy significantly correlated with instructor support
Ching et al., 2020, Hong Kong [10]	Self-awareness Coping strategy	Reflection on expectations vs. reality Peer/mentor networks for advice and emotional support Students’ reflective processes in adapting to clinic expectations	Self-efficacy, cognitive appraisal, coping mechanisms	Positive influence on resilience and coping among self-aware students, negative impact (self-blame) on low-resilience students
Curl et al., 2016, USA [26]	Pre-lab preparation Structured debriefing	Students actively engaged in role-playing scenarios followed by structured, reflective debriefings Case study questions completed before 45–90 min guided reflection post-simulation simulations to establish baseline knowledge	Self-efficacy, clinical reasoning, reflective practice Reduces cognitive load during simulations, enabling focus on skill application	Significantly improved competence, increased their confidence in technical skills and critical thinking, implied resilience via improved confidence Enhanced knowledge retention (higher HESI scores)
Dowling et al., 2021, USA [27]	Belongingness Student engagement Academic support	Creating an inclusive climate through equity training, supportive mentoring, holistic admissions, and early academic interventions	Self-efficacy, modeling inclusivity and equity, supportive relationships, reduction in isolation	Enhanced resilience, increased competence, motivation, satisfaction, retention
Fadana & Vember, 2021, South Africa [32]	Interpersonal support Effective communication Engagement (negative)	Poor interpersonal relationships and ineffective communication diminished support and engagement	Lack of modeling positive behaviors, reduced self-efficacy, increased anxiety	Reduced competence, (missed learning opportunities) Lower resilience (stress, isolation)
Gheisari et al., 2025, Iran [34]	Frequent and individualized feedback Structured reflection (checklist use)	Supervisors identified gaps and provided tailored feedback during handovers ISBAR checklist guided data interpretation, observation-based guidance, emotional and educational support	Enhanced self-efficacy through supportive feedback, structured modeling (ISBAR checklist adherence)	Improved clinical decision-making, increased self-efficacy and resilience to clinical stress
Hill et al., 2023, Ireland [36]	Reflective debriefing Peer observation	Structured reflection and facilitated discussions post-simulation, -observation, and -peer feedback	Self-efficacy, reflective practice, vicarious learning (observational learning)	Improved critical thinking, (competence) enhanced clinical judgment, increased confidence and adaptability (emotional resilience) through reflection and peer learning
Ireland, 2022, USA [28]	Reflective practice Role modeling Supportive faculty engagement	Encouraging self-assessment and growth mindset Faculty demonstrating EI in teaching and feedback, inclusive behavior	Self-efficacy, cognitive reframing, reflective learning	Strengthened self-awareness and adaptability Improved empathy and professional comportment
Jun & Lee, 2017, South Korea [38]	Ego-resiliency	Students’ ability to adapt and remain effective in stressful clinical situations	Ego-resiliency as a buffer against anxiety’s effects on cognitive functioning	Mediates the negative effects of anxiety on problem-solving competence
Labrague et al., 2025, Philippines [40]	Psychological resilience	Ability to bounce back from stress and adapt to clinical demands	Self-efficacy, adaptive coping	Higher resilience improves clinical adjustment and competence
Liang et al., 2019, Taiwan [31]	Peer support Mentorship Reflection	Shared experiences and coping strategies in group discussions Diary entries and debriefing on clinical challenges Non-evaluative guidance from faculty	Self-efficacy, social learning, modeling	Strengthened resilience through normalization of stressors Enhanced competence via self-regulated learning Increased confidence and clinical skill mastery
Ma et al., 2024, China [41]	Social support	Students’ ability to cope and adapt under stress	Social support foster resilience through self-efficacy and work readiness	Directly improves work readiness
Mehdipour-Rabori et al., 2021, Iran [35]	Formative and individualized feedback	Daily checklists, goal setting, repeated assessments	Enhance self-efficacy through competency-based learning	Improved skill mastery and stress management
O’Sullivan et al., 2021, Ireland [37]	Skill application Self-reflection	Students practiced and reflected decider skills in clinical and personal contexts	Self-efficacy, cognitive restructuring cognitive behavioral therapy/dialectical behavior therapy (CBT/DBT) principles	Improved emotional regulation, interpersonal interactions, and reduced stress
Park et al., 2022, South Korea [39]	Support Emotional expression reflection	Sharing experiences and writing poems helped emotional ventilation and peer empathy	Self-reflection, emotional awareness, cognitive reframing	Enhanced self-awareness, reduced stress, and enhanced resilience
Stacey et al., 2017, England [29]	Reflective discussion Mindfulness	Safe space, guided discussion of emotional regulation systems	Self-efficacy through positive reframing and distress tolerance	Enhanced resilience by reducing self-criticism and internalization of stress
Yu et al., 2021, China [42]	Self-efficacy	Student belief in ability to manage clinical tasks and challenges	Self-efficacy: belief influences behavior	Explains the effect between learning environment and competence
Zulu et al., 2021, South Africa [33]	Supportive relationships Structured clinical guidance	Professional nurses provided guidance and emotional support, clear goals, and regular feedback from tutors	Modeling, self-efficacy, purpose-driven coping	Enhanced competence through skill practice; resilience via emotional backing Strengthened clinical competence and adaptability

## Data Availability

No new data were created or analyzed in this study. Data sharing is not applicable to this article.

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
