# Peer review of "Building Resilience and Competence in Bachelor Nursing Students: A Narrative Review Based on Social Cognitive Theory"

_nursrep, 2025, doi:10.3390/nursrep15070253_

Round 1
Reviewer 1 Report
Comments and Suggestions for Authors
Please see the attached document

Comments on the Quality of English Language
Editing is required, some sentences are long.
Author Response
Comment 1: Dear authors, Thank you for submitting your review. It addresses a significant contemporary topic in nursing education that deserves attention. I have a few recommendations that could enhance your work further, and I encourage you to consider them as you continue to refine your review.
Response 1: We appreciate the reviewer`s recommendation.
Comment 2: Abstract
The authors provided a detailed abstract, which is commendable. However, including of information regarding the appraisals of the studies would enhance the overall quality of the abstract.
Response 2: Thank you for the constructive feedback. The following sentence was added in the abstract. (Page 1 Line 24-25)
“No formal study appraisal was conducted due to the narrative nature of the review.”
Comment 3: Introduction
Question 3. "What educational processes mediate the relationship between educator inputs and the development of competence and resilience?" This question is vague and not clearly formulated. To enhance clarity, it would be beneficial for the authors to specify the educational processes involved and detail the inputs from educators.
Response 3: We appraciate the valuable suggestion. We acknowledge that the original wording of Research Question 3 may appear vague without additional context. To address this, we have retained the original question but added a clarifying sentence immediately after the list of research questions in the Introduction section (Page 3 Line 99). We hope this addition improves clarity and helps readers better understand the conceptual scope of the question. (Page 3 Line 100-102)
“This sentence defines “educator inputs” as actions such as supervision, feedback, and role modelling, and “educational processes” as mechanisms including reflective practice, peer learning, and the development of self-efficacy.”
Comment 4: Materials and Methods
The authors provided a rationale for the design of their study, and the choice of databases is appropriate for the research conducted. Including the specific Boolean search strings would greatly improve the clarity of their methodology. Additionally, providing a detailed search strategy for one of the databases would allow readers to replicate the search more easily and further engage with the study's findings.
Response 4:
Thank you for your valuable feedback. The following changes were made to the revised manuscript (Page 3 Line 116-129)
“The search strategy combined four key concept areas using Boolean operators: (1) nursing students, (2) clinical education settings, (3) teaching strategies and educator behaviors, and (4) outcomes such as resilience and competence. The following search string was used in PsycINFO (Ovid): ("nursing students" or "undergraduate nursing" or "student nurse").mp. [mp=title, abstract, heading word, table of contents, key concepts, original title, tests & measures, mesh word] AND ("clinical education" or "clinical placement" or "nursing education").mp. [mp=title, abstract, heading word, table of contents, key concepts, original title, tests & measures, mesh word] AND ("teaching strategies" or "educator behaviors" or "instructional methods" or "clinical supervision").mp. [mp=title, abstract, heading word, table of contents, key concepts, original title, tests & measures, mesh word] AND ("resilience" or "psychological resilience" or "clinical competence" or "emotional safety" or "self-efficacy" or "emotional intelligence").mp. [mp=title, abstract, heading word, table of contents, key concepts, original title, tests & measures, mesh word].”
Comment 5: Results
Overall, the results included evidence from most of the included studies.
Response 5: Thank you
Comment 6: The PRISMA flow chart is a widely recognised tool used in reviews. It is recommended that the authors incorporate a PRISMA flow diagram to improve the transparency and clarity of the study selection process.
Response 6: Thank you for your helpful comment. While this is a narrative review and not a systematic review, we agree that transparency in the literature selection process is important. Therefore, we have included a PRISMA flow diagram as a supplementary file for clarity.
(Page 4 Line 157-158)
Comment 7: The authors included peer-reviewed articles published between 1 January 2000, and 2 May 2025, which is a good time frame for analysis. However, providing a clear rationale for these specific dates would enhance the study by allowing readers to evaluate any potential biases in the selection process.
Response 7: Thank you for this valuable suggestion. We selected the period from 1 January 2000 to 2 May 2025 to capture key developments in nursing education over the past 25 years. This timeframe encompasses major shifts in clinical teaching strategies such as the adoption of simulation-based learning, and an increased emphasis on clinical education and student preparedness in undergraduate nursing curricula. We have now included this rationale into the revised manuscript. (Page 4 Line 140-142)
“The specific time frame was selected to capture major shifts in nursing education over the past 25 years, including the integration of simulation, evolving clinical teaching strategies, and increased focus on competence in undergraduate clinical training.”
Comment 8: Discussion
The discussion section provides a clear theoretical foundation and effectively links the findings.
Response 8: Thank you for the kind feedback. We're pleased that the discussion came through clearly and that the theoretical foundation felt strong. We put careful thought into using Bandura’s Social Cognitive Theory to tie the findings together and make their relevance to nursing education as clear and meaningful as possible.
Comment 9: The authors included SCT in discussion section, which is commendable. The application of SCT could benefit from a more analytical approach rather than remaining predominantly descriptive. For instance, while self-efficacy is rightly emphasised, a deeper exploration into how and why certain strategies vary in effectiveness across different contexts or student populations could provide valuable insights.
Response 9: Thank you for this insightful comment. We appreciate your acknowledgment of our integration of Social Cognitive Theory (SCT) and agree that a more analytical application would enhance the discussion. In response, we have revised the discussion to deepen the theoretical engagement with SCT particularly by exploring how and why specific strategies (e.g., simulation, peer support, feedback) may vary in their effectiveness across different student populations, learning environments, and cultural contexts.
Relevant changes have been made and highlighted throughout discussion section.
Page 19 Line 360-369
Page 19-20 Line 378-389
Page 20 Line 397-410
Page 20-21 Line 421-437
Page 21 Line 446-454
Page 21 Line 464-470
Page 21-22 Line 482-494
Comment 10: The authors offered a comprehensive overview of the limitations, which provides valuable insight for further improvement.
Response 10: Thank you for your positive feedback. We appreciate your recognition of the limitations section and its role in guiding future research.
Comment 11: Writing
Some sentences were long and may benefit from being shortened for better clarity. By breaking up longer sentences, the authors can improve understanding. Authors are recommended to properly edit the manuscript to improve its overall quality.
Response 11: The manuscript was proofread and reviewed for clarity, flow, and grammatical accuracy.
Reviewer 2 Report
Comments and Suggestions for Authors
Dear authors,
I had the honor of reading your paper. I hope it will be accepted with some small suggestions.
As your work relates to a particular theory, I think that adding "Social Cognitive Theory" to the title would make the theoretical framework more transparent, for example:
"A Narrative Review Based on Social Cognitive Theory".
The method is perhaps the weakest part of the work:
A specific display of search phrases is missing, e.g. "((nursing students) AND (clinical placement)) AND (resilience OR competence)", which would enable replicability.
Also a display of the selection logic (eg PRISMA diagram, although not mandatory in narrative reviews) would make the selection much more transparent.
It was not stated who performed the selection and whether there was double coding or independent review, as well as how possible inconsistencies were resolved.
The authors could have made at least a descriptive assessment of the strength of the evidence (eg "most studies were small-N, self-report based", etc.). They stated it in limitations, but it would be really effective to connect the power of the evidence with the facts themselves.
In chapter 3.3. I think it would be useful to list which strategies were used in which papers.
They could have mentioned the possibilities of low-budget simulations (low-fidelity), which was mentioned only briefly.
Consider as a possibility overview tables in the supplement that would contain key facts for the included studies. These tables would further enhance the transparency of the results and be useful to readers who want to see the characteristics of the included studies.
Best,
Author Response
Comment 1: Dear authors,
I had the honor of reading your paper. I hope it will be accepted with some small suggestions.
As your work relates to a particular theory, I think that adding "Social Cognitive Theory" to the title would make the theoretical framework more transparent, for example:
"A Narrative Review Based on Social Cognitive Theory".
Response 1: We appreciate the thoughtful feedback and support for the manuscript.
We agree that explicitly referencing the theoretical framework in the title improves clarity and transparency. We have revised the title to:
Page 1 Line 2
“Building Resilience and Competence in Bachelor Nursing
Students: A Narrative Review Based on Social Cognitive Theory”
Comment 2: The method is perhaps the weakest part of the work:
A specific display of search phrases is missing, e.g. "((nursing students) AND (clinical placement)) AND (resilience OR competence)", which would enable replicability.
Response 2: Thank you for the helpful comment. We have made changes in the method section as suggested. (Page 3, Line 117-130)
Comment 3: Also a display of the selection logic (eg PRISMA diagram, although not mandatory in narrative reviews) would make the selection much more transparent.
Response 3: Thank you for your helpful comment. While this is a narrative review and not a systematic review, we agree that transparency in the literature selection process is important. Therefore, we have included a PRISMA flow diagram as a supplementary file for clarity. (Page 4 Line 157-158)
Comment 4: It was not stated who performed the selection and whether there was double coding or independent review, as well as how possible inconsistencies were resolved.
Response 4: Thank you for the comment. We have updated the Methods section to clarify that one reviewer independently screened all records for inclusion. Any discrepancies were discussed and resolved within the review team through group consensus. This revision enhances the transparency of our selection process. (Page 4 Line 153-155)
“Titles and abstracts were screened for relevance by one reviewer (SK), followed by full-text screening of eligible articles (SK). The extracted data and disagreements were discussed and resolved through consensus within the review team”
Comment 5: The authors could have made at least a descriptive assessment of the strength of the evidence (eg "most studies were small-N, self-report based", etc.). They stated it in limitations, but it would be really effective to connect the power of the evidence with the facts themselves
Response 5: Thank you for this thoughtful comment. We agree that connecting the descriptive strength of the evidence to the studies presented in Table 1 adds clarity and contextual rigor. In response, we have added a paragraph in the Results section summarizing the methodological characteristics of the 19 included studies. This helps to qualify the strength of the evidence presented and complements the limitations already discussed (Page 5, Line 185-189)
“The majority employed descriptive, cross-sectional, or quasi-experimental designs, and many relied on self-reported outcomes from students or faculty. Sample sizes varied considerably, but several studies involved relatively small cohorts especially in qualitative or mixed-methods designs. Few studies used validated clinical assessments”
Comment 6: In chapter 3.3. I think it would be useful to list which strategies were used in which papers.
Response 6: Thank you for your helpful suggestion. We agree that mapping specific strategies to individual studies could add analytical depth. However, to preserve the clarity and flow of our narrative synthesis, we chose to present findings thematically rather than as a detailed matrix. For readers wishing to trace strategy-paper links, this can be done through the existing in-text citations in section 3.3. We appreciate your recommendation and will consider incorporating a mapping table in future publications or as supplementary material if appropriate.
Comment 7: They could have mentioned the possibilities of low-budget simulations (low-fidelity), which was mentioned only briefly.
Response 7: Thank you for your insightful comment. We agree that low-budget or low-fidelity simulations represent an important and practical alternative, especially in low-resource settings. To address this, we have slightly expanded the relevant paragraph to more clearly emphasize the potential of low-fidelity approaches as scalable, cost-effective solutions aligned with Social Cognitive Theory. (Page 19 Line 359-364)
“However, scalable adaptations such as low-fidelity simulations using basic mannequins, role-play, or scenario-based discussions offer viable, cost-effective alternatives for skill-building in low-resource environments [42]. These methods preserve the core SCT principle of mastery learning by enabling repeated, guided practice, even when high-tech resources are unavailable.”